# Characteristics of Internal Solitary Waves in the Timor Sea Observed by SAR Satellite

**Yunxiang Zhang** [1], **Mei Hong** [1], **Yongchui Zhang** [1,*], **Xiaojiang Zhang** [1], **Jiehua Cai** [1], **Tengfei Xu** [2,3] and **Zilong Guo** [1]

1   College of Meteorology and Oceanography, National University of Defense Technology, Changsha 410073, China; zhangyunxiang@nudt.edu.cn (Y.Z.); hongmei17@nudt.edu.cn (M.H.); zhangXiaojiang19@nudt.edu.cn (X.Z.); caijiehua@whu.edu.cn (J.C.); gzlong@nudt.edu.cn (Z.G.)
2   Key Laboratory of Marine Science and Numerical Modeling, First Institute of Oceanography, Ministry of Natural Resources, Qingdao 266061, China; xutengfei@fio.org.cn
3   Laboratory for Regional Oceanography and Numerical Modeling, Pilot National Laboratory for Marine Science and Technology, Qingdao 266237, China
*   Correspondence: zyc@nudt.edu.cn

**Abstract:** Internal solitary waves (ISWs) with features such as large amplitude, short period, and fast speed have great influence on underwater thermohaline structure, nutrient transport, and acoustic signal propagation. The characteristics of ISWs in hotspot areas have been revealed by satellite images combined with mooring observation. However, the ISWs in the Timor Sea, which is located in the outflow of the ITF, have not been studied yet and the characteristics are unrevealed. In this study, by employing the Synthetic Aperture Radar (SAR) images taken by the Sentinel-1 satellite from 2017 to 2022, the temporal and spatial distribution characteristics of ISWs in the Timor Sea are analyzed. The results show that most of ISWs appear in Bonaparte basin and its vicinity. The average wavelength of the ISWs is 248 m, and most of the wave lengths are less than 400 m. The peak line of ISWs is longer in deeper water. The underwater structures of two typical ISWs are reconstructed based on the Korteweg–de Vries (KdV) equation combined with mooring observation. This shows that, compared with the two-layer model, the continuous layered model is more suitable for reconstructing the underwater structures of ISWs. Further analysis shows that both the rough topography and the spring-neap tides contribute to the generation of ISWs in the Timor Sea. This study fills a gap in knowledge of ISWs in regional seas, such as the Timor Sea.

**Keywords:** internal solitary waves; SAR images; Korteweg–de Vries equation; Timor Sea

## 1. Introduction

Internal solitary waves (ISWs) are hump-shaped, large-amplitude waves and propagate horizontally within the ocean [1]. Typically, ISWs consist of a single isolated wave of elevation or depression, depending on the background state [2]. ISWs are of great importance to various scientific issues, including sound propagation [3], pollutant transport [4], and primary productivity [5]. ISWs carry a lot of energy and can cause strong density displacement, which is serious threat to underwater structures and submarines [6–8]. For example, studies show that the Indonesian navy submarine KRI Nanggala 402 is likely to have had an accident that was affected by ISWs [9,10].

At present, the main means of ISW observation include direct measurement in the field and indirect measurement using remote sensing observation. Field measurement can observe the ISWs' underwater structures. However, it is always expensive and only one or more observation sites can be set. Chen et al. [11] and Huang et al. [12] studied the dynamic and statistical features of the ISWs in the northern part of the South China Sea using mooring data. Yang et al. [13] revealed the complex behaviors of ISWs in the

Andaman Sea using data collected over a nearly 22-month-long observation period, completed by two moorings. By contrast, remote sensing observation can obtain the regional distribution of ISWs with fine spatial imaging ability. Among the remote sensing means, the spaceborne synthetic aperture radar (SAR) is an effective piece of equipment for observing ISWs through sea surface roughness, through the Bragg scattering mechanism [14]. Brighter and darker stripes in SAR images correspond to rougher and smoother areas on the sea surface, respectively. According to their vertical structures, ISWs can be divided into two types: elevation internal waves and depression internal waves [15], which correspond to convergence and divergence on the sea surface, respectively. For the depression ISWs, the rough capillary waves will be generated on the seawater surface corresponding to the position in front of the depression internal wave along its propagation direction, and the seawater at the corresponding position is brighter when observed from SAR images. Thus, the surface water corresponding to the position behind ISWs is smooth due to divergence and is dark from the SAR image, which will form stripes that are bright first and then dark along propagation direction. For the elevation ISWs, stripes are dark first and then bright on SAR images. When ISWs propagate from deep to shallow, the polarity conversion can be observed in satellite SAR images [16]. Liu et al. [17] calculated the phase speeds of ISWs in the South China Sea using SAR image pairs from the Envisat and ERS-2 tandem satellites. Adi et al. [18] constructed the vertical structure of ISW packets in the Maluku Sea and discussed the energy sources of ISWs in the local area. Kudryavtsev et al. [19] made statistics on spatial properties of observed ISWs trains from 354 ENVISAT Advanced Synthetic Aperture Radar (ASAR). However, only the surface characteristics can be determined from the SAR images. Since each method has specific advantages and disadvantages, combining methods can be used to verify each individual method.

It is known that ISWs are generated when a strong tidal current flows through rough topography such as a continental slope, seamount, and shallow shore. The generation process of ISWs is usually explained as the lee wave mechanism [20,21] and the nonlinear internal tide mechanism [22,23]. According to the lee wave mechanism, ISWs are formed by tidal-topographic nonlinear interaction. When the sea water passes through the ridge at low tide, it will form a depression internal wave. When the ebb tide turns into the flood tide, the depression internal wave steepens nonlinearly, and evolves into ordered ISW packets, which propagate along the direction of the tidal current [24]. For the nonlinear internal tide mechanism, internal tides spawn ISWs in three steps: initial generation of a front due to topographic blocking, nonlinear steepening of the front, and formation of a rank-order ISW packet under the effects of nonlinearity and dispersion [25]. However, the ISW generation mechanisms are suitable for different situations. For example, the lee wave mechanism could reasonably explain the generation of ISWs at an underwater sill or a bank, but it cannot explain ISWs' generation at the shelf break, which is more consistent with the nonlinear internal tide mechanism [26].

ISWs are frequently found in most of the world's oceans, such as the Andaman Sea, the South China Sea, and the Gulf of Oman. Tensubam et al. [27] used multiple image comparison and tidal time period methods to estimate the phase speed of ISWs. It was found that water depth and monthly stratification play a vital role in controlling the phase speed of the ISWs in the Andaman Sea. By using the Moderate Resolution Imaging Spectroradiometer (MODIS) images of Terra/Aqua satellites from 2014 to 2016, combined with mooring array observations, Yang et al. [28] revealed the three-dimensional structures of ISWs in the northern South China Sea. Zhang et al. [29] revealed a cascade process from ISWs to turbulent mixing via high-frequency internal waves near the maximum local buoyancy frequency in the deep water of the northern South China Sea by high-resolution mooring observations. Koohestani et al. [30] used a combination of multiple data sources to study the characteristics of ISWs in the Gulf of Oman. Beyond that, due to the complex islands and rough topography, the outflow of the Indonesian Transport Flow (ITF) is a hotspot for ISWs. Adi et al. [31] observed an ISW with an amplitude of more than 90 m around 8 km from the western coast of Lombok Island by moorings, and found that it

was dominated by an M2 tidal component. Karang et al. [32] extracted the temporal and spatial distribution of ISW propagation in Lombok from the described Landsat 14 image. The result shows that the average ISWs' phase velocity was 2.05 m/s, with the direction of propagation heading north at an average angle of 19.08°. However, previous research on ISWs focuses on the Lombok Strait and Bali waters. Though the Timor Sea is a sea area with a high frequency of internal waves [33], it attracts little research.

In order to fill the knowledge gap of ISWs in the Timor Sea, the temporal and spatial distribution characteristics of ISWs are revealed by the SAR images, and the underwater structures of ISWs are reconstructed, combined with the theory and observations in this study. Notably different from the statistical research based only on remote sensing images, the study illustrates the accuracy of different theoretical models with the observation data of moorings. The study is organized as follows. In Section 2, the remote sensing image data and mooring data, as well as the methods, are introduced. In Section 3, the statistical characteristics of ISWs in the Timor Sea based on SAR images are revealed. In Section 4, the underwater structures of ISWs are reconstructed and compared with the mooring data. In Section 5, the generation mechanism of ISWs is analyzed based on the local tidal and topographic features. The last section is the conclusion.

## 2. Materials and Methods

### 2.1. Data Introduction

#### 2.1.1. Satellite Images

The products of Copernicus Sentinel Data processed by the European Space Agency (ESA) were downloaded from https://search.asf.alaska.edu (accessed on 17 December 2022). The satellite images captured were by the mission of Sentinel 1-A, the beam mode was interferometric wide swath, product type was ground-range detected, high-resolution processing level 1, and polarization type was VV. To obtain accurate geographic information from the SAR image, the image was preprocessed by geocoding and geometric correction. The SAR image coverage is indicated in black dotted box in Figure 1. There are 345 SAR images in total during the six years from 2017 to 2022.

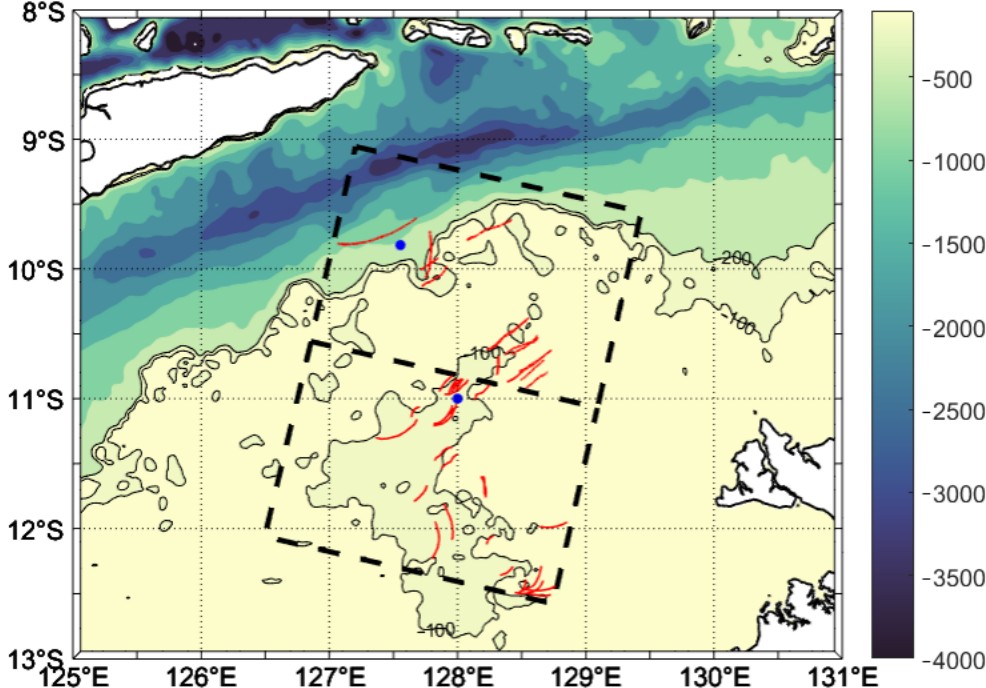

**Figure 1.** Topography and SAR frames. The shaded color is the bathymetry with 100 m and 200 m depth of black lines. Two dashed rectangular boxes are the two SAR frames. Red arcs are the leading wavefronts detected by the ISWs. Two blue dots are the mooring sites.

2.1.2. Mooring Data

Two moorings in the Timor Sea as part of Australia's Integrated Marine Observing System (IMOS) were collected, which are marked with blue dots in Figure 1. One mooring is located at 127.558E, 9.828S in 465 m of water on the south continental slope of Timor Trough. This trough is about 150 km wide with a maximum depth of 3000 m. To the south of the mooring is the Australian North-West shelf, which is approximately 400 km wide and 100 m deep. The mooring is equipped with thermistors at 20 m, 30 m, 40 m, 50 m, 60 m, 70 m, 80 m, 100 m, 115 m, 130 m, 155 m, 180 m, 200 m, 250 m, 300 m, 350 m, 400 m, and 460 m depth, which sample every 2 min or less. Another mooring is located at 128.008°E, 11.008°S at a depth of 145 m on the Australian North-West shelf. The geological area of the continental shelf is called Bonaparte basin. The mooring is located in the center of a small canyon with a northeast-southwest trend and is about 20 km wide and 50 m deep. The mooring is equipped with thermistors at 20 m, 30 m, 40 m, 50 m, 60 m, 70 m, 80 m, 100 m, 120 m, and 140 m depth, which sample every 2 min or less.

*2.2. Identification of ISWs from SAR Images*

In order to identify the ISWs' features on SAR images, the method of Da Silva et al. [34] was employed to measure the relative intensity distribution along the wave propagation direction on SAR images. The relative intensity is defined as

$$\frac{\delta I}{I_0} = \frac{I - I_0}{I_0} \tag{1}$$

where $I$ is the average value of reflection intensity on multiple pixel lines along the propagation direction of wave packet, and $I_0$ is the intensity of the image background, which comes from the average value of the reflection intensity of a certain area far away from the ISWs' field. In order to obtain meaningful intensity measurement and reduce speckle, it is necessary to take the average value as at least 500 pixels [34]. The condition is satisfied by averaging the reflection intensities on at least 10 pixel lines consisting of more than 60 pixels along the ISWs' propagation direction.

Although the half-wave width of ISWs can be theoretically determined by measuring the distance of between two adjacent peaks on the backscattering intensity profile curve, the actual signal is often disturbed by other interfering signals and noises. Thus, the signal analysis methods are needed to extract the signal of ISWs. Empirical mode decomposition (EMD) is a common method for retrieving the parameters of ISWs [35–37]. The EMD method does not need to preset any basis functions, so it is suitable for processing nonlinear and non-stationary signals [38]. Compared with the other signal processing methods, such as wavelet decomposition or Fourier transform, EMD has a higher resolution, strong adaptability, and easy implementation.

The EMD method assumes that the series is the superposition of many intrinsic mode functions (IMF) and residuals. IMF represents the characteristics of this series on different scales. The process of the EMD method given by Huang [38,39] is as follows:

1. For series $x(t)$, find all of the local minima and maxima of it, then construct the upper and envelope of the series by cubic spline interpolation.
2. The average value of the upper and lower envelopes is recorded as the average envelope. The series is subtracted from the average envelope to obtain a new series $x'(t)$.
3. Check whether the new series $x'(t)$ meets the definition of IMF. The IMF needs to meet two conditions: first, the number of extreme points is equal to or at most one different from the number of crossing zeros; second, the average value of the upper envelope composed of local maxima and the lower envelope composed of local minima is zero. If the above conditions are met, the new series is the IMF. Otherwise, repeat Step 1 and Step 2 for $x'(t)$ until the conditions are met.

4.  After obtaining the IMF using the above steps, the IMF is subtracted from the original sequence $x(t)$ to obtain a new sequence $x_1(t)$. Repeat Steps 1 to 3 for $x_1(t)$ to obtain next IMF until the series cannot be decomposed any more.

The relative energy of each model can be estimated by calculating the variance of each model obtained by EMD and normalizing it. The normalized variance of each mode is

$$\sigma_i = \frac{var_i^2}{\sum_{i=1}^{m} var_i^2} \qquad (2)$$

where $m$ represents the number of decomposed modes and $var_i^2$ stands for the variance of the $i$-th IMF obtained by EMD. Since the energy of the ISWs is large, the mode with the largest normalized variance $\sigma_i$ represents the internal wave signal, and its curve shape should be similar to that of the backscattering intensity.

### 2.3. Reconstruction of the Underwater Structure of ISWs

There are a lot of internal wave propagation models describing the characteristics of ISWs depending on the relationship between the horizontal scale of ISWs and the water depth. If the wave length of internal waves is much less than the water depth, the Benjamin Ono equation is suitable [40]. If the wavelength of internal waves is close to the water depth, the Joseph Kubota equation [41] is appropriate. However, if the horizontal wavelengths of ISWs are much larger than the water depth, such as in the Timor Sea, the KdV equation [42], which only keeps the first order approximations of nonlinearity and dispersion, is applicable.

Based on the KdV theory, the common models used for ISW reconstruction are the continuous layered model and the two-layer model. Generally speaking, if the background density profile is accurate, the amplitude estimation from SAR images using a continuous stratified fluid model is more accurate than that obtained by using a two-layer ocean model [43]. However, the two-layer model has its advantages, such as clear physical meanings and low computational complexity. The reconstruction process under the two models is introduced below.

### 2.3.1. KdV Equation in Continuously Stratified Liquid

The ISW amplitude estimation method using a classical KdV equation in continuous layered ocean was proposed by Small et al. [44]. Here, we briefly describe the method. In a stratified fluid, the amplitude of ISW η satisfies the Korteweg–de Vries (KdV) equation

$$\frac{\partial \eta}{\partial t} + c\frac{\partial \eta}{\partial x} + \alpha\eta\frac{\partial \eta}{\partial x} + \beta\frac{\partial^3 \eta}{\partial x^3} = 0 \qquad (3)$$

where $x$ is the horizontal coordinate, $t$ is time, and $c$ is the linear phase speed. A classical solution to above equation is

$$\eta = \eta_0 sech^2\left(\frac{x - Vt}{L}\right) \qquad (4)$$

where $\eta_0$ is the ISW amplitude, $V = c + \frac{\alpha\eta_0}{3}$ is the soliton velocity, and $L = \sqrt{\frac{12\beta}{\alpha\eta_0}}$ is defined as the characteristic half width of the soliton. In the continuously stratified water of depth $H$, the nonlinear coefficient $\alpha$ and dispersive coefficient $\beta$ are written as [45]

$$\alpha = \frac{3c \int_{-H}^{0}(d\varphi/dz)^3 dz}{2 \int_{-H}^{0}(d\varphi/dz)^2 dz} \qquad (5)$$

$$\beta = \frac{c \int_{-H}^{0} \varphi^3 dz}{2 \int_{-H}^{0}(d\varphi/dz)^2 dz} \qquad (6)$$

where $\varphi(z)$ is the first-mode vertical eigenfunction of the wave. The continuous ocean can be separated by taking the vertical structure of a discrete set of linear modes. Since the two ISW cases selected are shallow water long waves, they satisfy the weak nonlinear condition [45]. Thus $\varphi(z)$ is given by the Sturm–Liouville equation

$$\frac{d^2\varphi(z)}{dz^2} + \frac{N^2(z)}{c^2}\varphi(z) = 0 \tag{7}$$

under the boundary conditions of $\varphi(-H) = \varphi(0) = 0$. $N(z)$ is the buoyancy frequency defined by

$$N^2(z) = -\frac{g}{\rho}\frac{\partial\rho}{\partial z} \tag{8}$$

where $g$ is the acceleration of gravity, and $\rho$ is the potential density. For KdV solitons, a simple relationship has been found between the half width $L$, and the distance $D$ between the center of bright and dark stripes in the SAR image [44].

$$D = 1.32L \tag{9}$$

The ISW amplitude $\eta_0$ could be derived from the ISW band with a bright-dark pattern in the SAR image:

$$\eta_0 = \frac{12\beta}{\alpha L^2} = \frac{12\beta}{\alpha\left(\frac{D}{1.32}\right)^2} \tag{10}$$

2.3.2. KdV Equation in the Two-Layer Liquid

If there is a stable density thermocline in the ocean, the two-layer fluid hypothesis can be used to simplify the reconstruction process. For a two-layer system with rigid cover and no background flow, the following equation can be obtained in Boussinesq approximation

$$c = \left(\frac{g\Delta\rho h_1 h_2}{\rho_0(h_1 + h_2)}\right)^{\frac{1}{2}} \tag{11}$$

$$\alpha = \frac{3}{2}c\frac{h_1 - h_2}{h_1 h_2} \tag{12}$$

$$\beta = \frac{c}{6}h_1 h_2 \tag{13}$$

where $h_1$ is the thickness of the upper water layer and $h_2$ is the thickness of the lower water layer. For a depression (elevation) wave, $h_1$ is smaller (larger) than $h_2$. In the two-layer model, the buoyancy frequency is zero except for the interface with infinite buoyancy frequency. Therefore, in order to represent the real ocean through two-layer approximation, this study chooses the depth of the maximum buoyancy frequency $N_{max}$ as the interface depth.

The relative layer density difference is

$$\frac{\Delta\rho}{\rho_0} = \frac{\rho_1 - \rho_2}{(\rho_1 + \rho_2)/2} \tag{14}$$

where $\rho_1$ ($\rho_2$) is the average density of the upper layer (lower layer) and is calculated as follows:

$$\rho_1 = \frac{\int_0^{h_1}\rho(z)dz}{h_1} \tag{15}$$

$$\rho_2 = \frac{\int_{h_1}^{h_1+h_2}\rho(z)dz}{h_2} \tag{16}$$

The amplitude can also be calculated by Equation (10).

### 2.3.3. Flow Field and Energy of ISWs

The vertical velocity is regarded as the partial derivative of isopycnal displacement with respect to time, thus

$$w = \frac{\partial \eta}{\partial t} \tag{17}$$

Combined with the continuity equation of incompressible flow $\frac{\partial u}{\partial x} = -\frac{\partial w}{\partial z}$, the horizontal current velocity induced by ISW is

$$u = -\eta_0 V \frac{d\varphi}{dz} sech^2 \left( \frac{V(t_0 - t)}{L} \right) \tag{18}$$

The density disturbance generated by ISWs and the current field calculated by Equations (17) and (18) are used to estimate the energy property of an ISW. The energy of ISWs consists of available potential energy (APE) and kinetic energy (KE). By integrating the vertical and horizontal ranges of ISWs, the APE density per unit volume is calculated as follows [46]

$$APE = \int_{z-\eta}^{z} g\left[\rho(z) - \rho_r\left(z'\right)\right] dz' \tag{19}$$

where $\rho_r(z')$ is the reference density at depth $z'$, which is initial density without solitary wave disturbance. The KE density per unit volume of ISW is calculated as

$$KE = \frac{1}{2}\rho\left(u^2 + w^2\right) \tag{20}$$

## 3. Results

### 3.1. The Statistical Characteristics of ISWs in Space and Time

There are a total of 345 SAR images collected from 2017 to 2022 in the Timor Sea. Among them, a total of 44 ISW trains that appeared on 20 SAR images are identified (Figure 1). Comparing these with the other regional seas, such as the Laptev Sea [19] (91 ISW trains on 354 ENVISAT ASAR images) and the White Sea [47] (516 ISWs on 282 SAR images), the ISWs are less identified in the Timor Sea. Even so, it can be found that most ISWs occur in the continental shelf area, while only a few ISWs appear in the southern edge of Timor Trench, at the junction with the continental shelf. Taking the 100 m isobath as the edge of Bonaparte basin, it can be found that most of the ISWs are distributed in or along the Bonaparte basin, especially the northern, northeast, and southeast edges of the basin.

Figure 2a shows the relative observation frequency of ISWs. It can be found that the maximum frequency is near (128°E, 11.3°N), which is the highest frequency of ISWs appearing. Generally, a single ISW packet usually contains 2–3 (maximum of seven) continuous solitons. As shown in Figure 2b, the main feature of ISWs in the Timor Sea is short wavelength, which is mostly less than 400 m, and the average wavelength is 248 m. The ISWs with the maximum wavelength appear at (127.8°E, 9.8°S), which is located near the 200 m isobath, and its wavelength exceeds 1 km. The average crest line length of ISW trains is about 24.2 km (Figure 2c). The longest wavefront, with a length of 74.0 km, is found at the edge of the northern continental shelf, which spreads from the deep sea to the continental shelf area and heads southward. Normally, the crest line of ISWs in the deeper area is longer than that in the shallow one, which reveals that the continental shelf may cause more energy dissipation and crest line breakage of ISWs. Figure 2d is the spatial distribution of ISWs' propagation directions. It can be seen that most ISWs propagate southeast or northwest, which is possibly influenced by the local internal tide current direction.

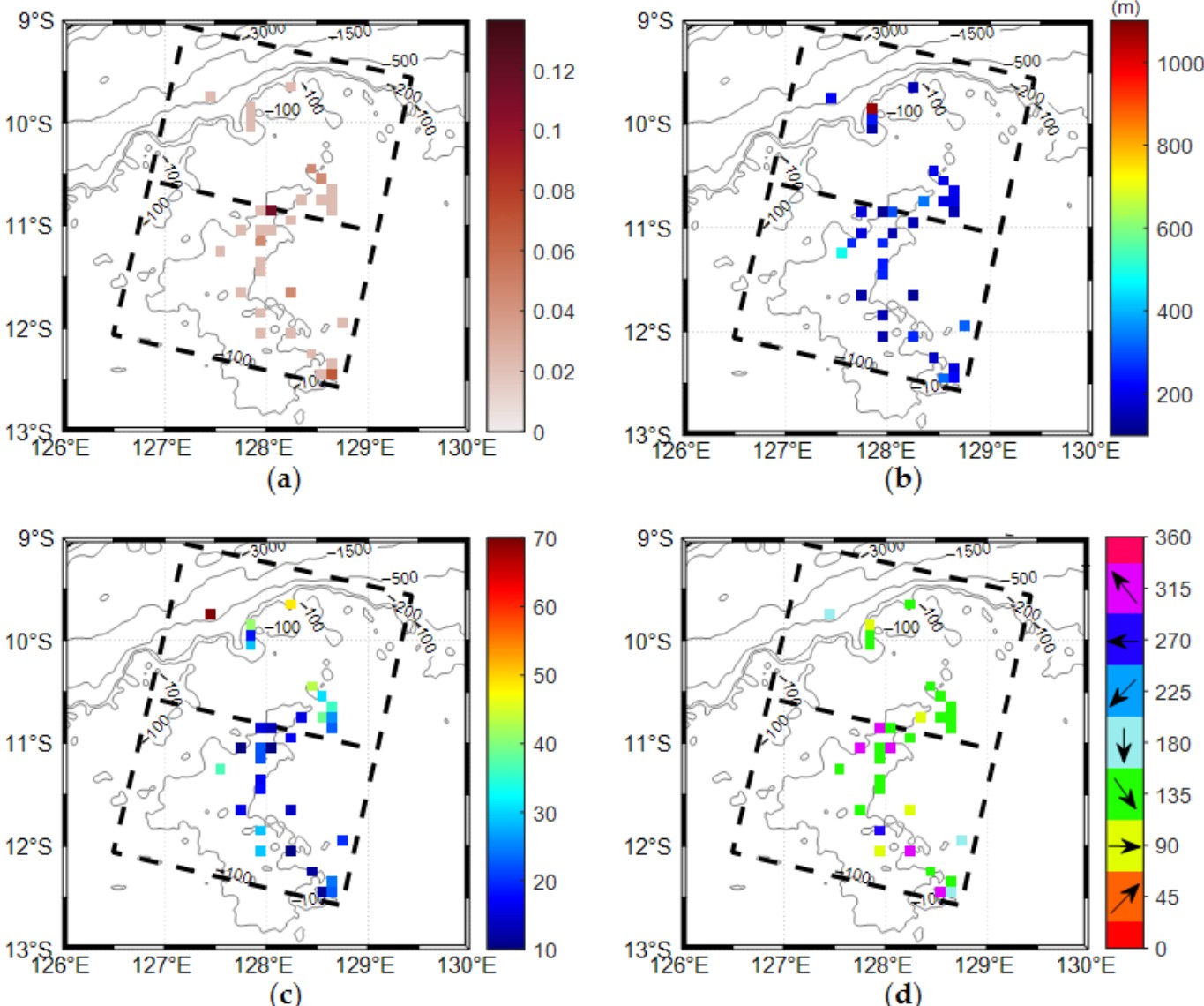

**Figure 2.** Spatial distribution of ISW parameters in the Timor Sea. (**a**) Observational frequency. (**b**) Wavelength of the leading wave in the packet [m]. (**c**) Crest lengths [km]. (**d**) Direction of ISWs' propagation.

The monthly distribution of ISWs are shown in Figure 3. It is shown that the number of ISWs in spring (13 ISWs) and autumn (20 ISWs) is more than in summer (four ISWs) and winter (seven ISWs). Note that the seasonal variation of the number of ISWs in the Timor Sea can be affected by the atmospheric conditions. Brandt et al. [48] pointed out that the ISWs' signal in remote sensing images is closely related to the wind speed, and it is more likely to be detected when the wind speed is low. When the wind speed is larger than 5 m/s, the image contrast for ISWs within the C band is less than 20%, which is difficult to observe visually. Figure 3 shows that the average wind speed in spring is 4.46 m/s and in autumn it is less than 4 m/s, while the average wind speed in other seasons is much higher than 5 m/s, which is in favor of observing ISWs in remote sensing images in spring and autumn. In addition, radar observation direction, precipitation, and variation of the depth of pycnocline will also affect remote sensing imaging. Therefore, the statistical results obtained from remote sensing images are limited to the true seasonal distribution of ISWs in the Timor Sea.

### 3.2. Case Study

3.2.1. Surface Characteristics of ISWs

In order to study the underwater structural characteristics of ISWs in the Timor Sea and verify the accuracy of the method of reconstructing ISWs' underwater structure through remote sensing images, two SAR images containing ISWs passing through the moorings were selected to validate. Figure 4a is a SAR image taken at 21:04:30 on 29 July 2017. It shows an ISW train propagating eastward and containing two significant crests, and is named ISW0729. The length of wave crest line is nearly 50 km. The image shows that the wave train has passed the mooring M1 at the time of image shooting, and the distance between them is about 25 km. Figure 4b is a partially enlarged SAR image, which was taken at 21:04:52 on 26 April 2017. Several ISWs with short peak lines can be clearly observed. The center of the wave train passing through the mooring M2 is approximately located at (127.95°E, 10.92°S), and its crest line length is only about 21 km. At the time of image shooting, the wave train has not reached M2, and the leading soliton of the train is about 7.8 km away from M2.

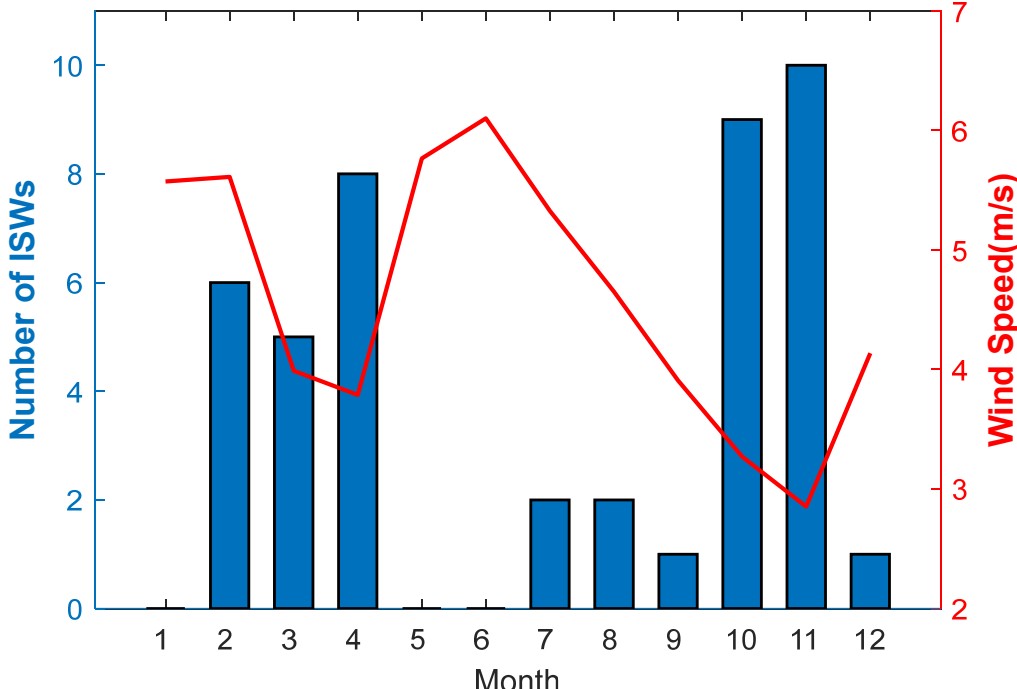

**Figure 3.** Monthly distribution of ISWs and wind speed. The blue bar graph indicates the number of ISWs, and the red line indicates the monthly average wind speed (m/s).

The wave signal before and after EMD is shown in Figure 5. For ISW0729, the normalized variance of the signal of mode 3 is the largest. Based on the distance between adjacent extreme values of the mode 3 signal, it can be determined that the average distance between the bright interval and the dark interval is 1420 m, thus the half wave width of ISW0729 could be calculated as 1076 m according to Equation (9). For the ISW0424, the normalized variance of the signal of mode 2 is the largest, and the average distance between the bright interval and the dark interval obtained from mode 2 is 230 m, thus the half wave width is 205 m. Considering that the image resolution is 30 m, the estimation error of half wave length is ±15 m.

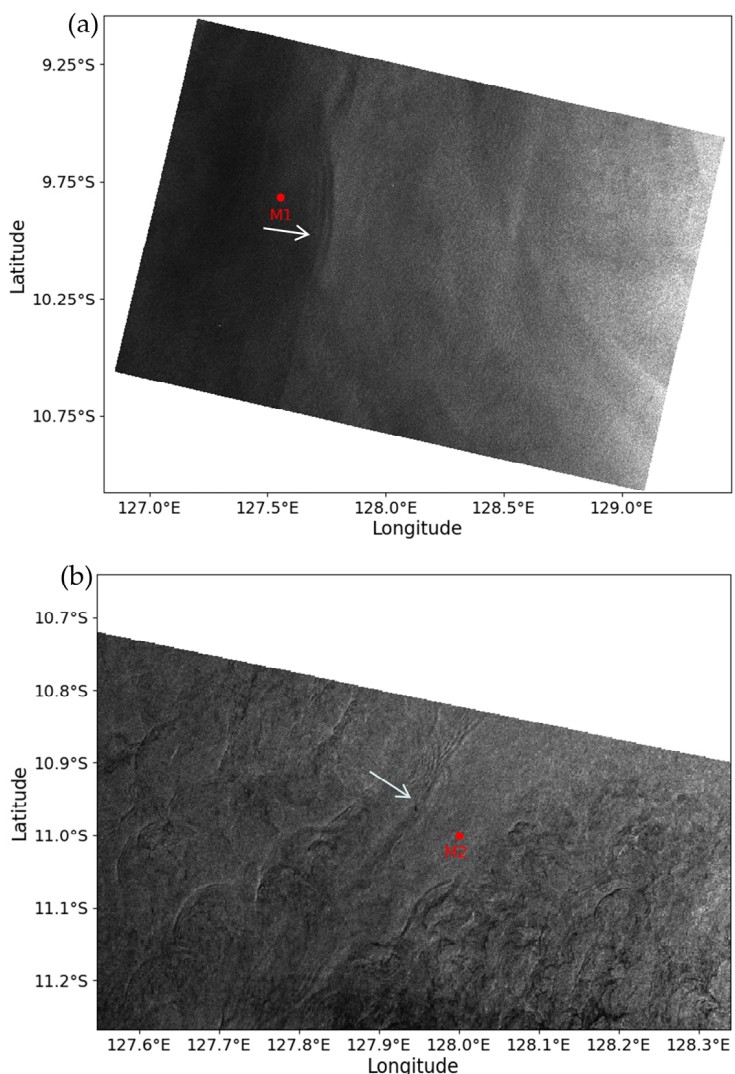

**Figure 4.** Sentinel 1-A satellite image with ISW cases taken at (**a**) 29 July 2017, (**b**). 24 April 2017.

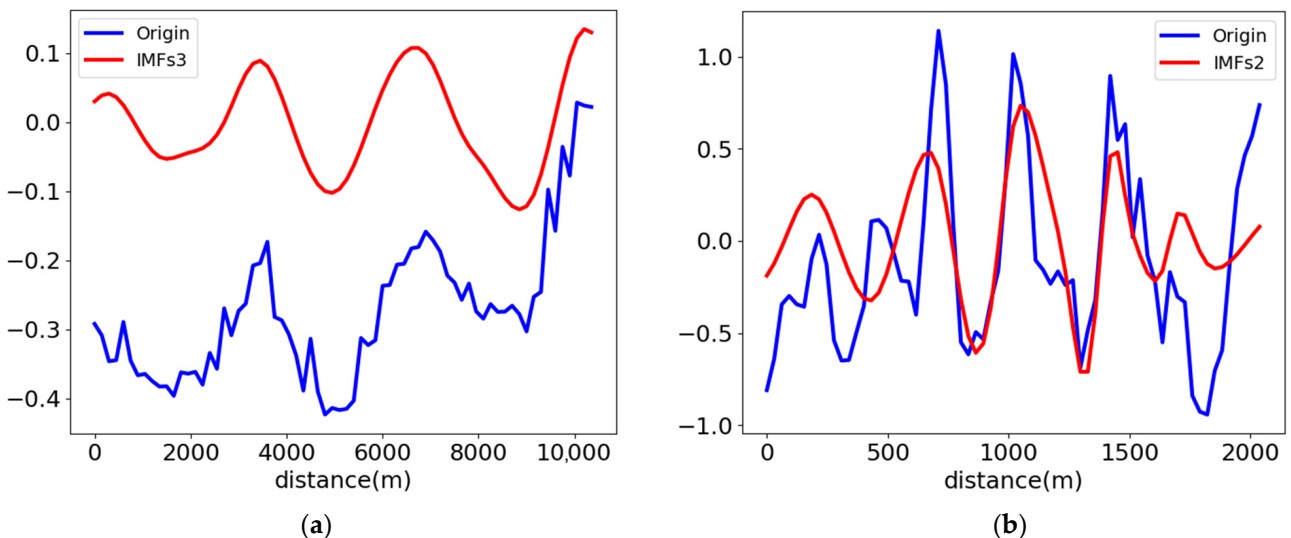

(**a**)                                                                                            (**b**)

**Figure 5.** Distances between bright and dark stripes by empirical mode decomposition (EMD) for (**a**) ISW0729 and (**b**) ISW0424. The blue line is the original backscattering intensity profile data series and the red line is the IMF with largest normalized variance obtained by EMD.

### 3.2.2. Underwater Structures of ISWs

Based on the reanalysis data products of monthly average temperature and salinity in the framework of the Copernicus Marine Environment Monitoring Service (CMEMS), the vertical density profile and buoyancy frequency $N$ in the position of the ISWs can be calculated, and then the eigenfunction $\varphi(z)$ of the first mode can be obtained by Equation (7) using the classical Thomson–Haskell method [49]. Figure 6a,c show the Brunt–Väisälä frequency and eigenfunction of ISW0729, respectively. It can be found that the maximum value of the Brunt–Väisälä frequency is located at about 90 m depth. However, the intrinsic function $\varphi(z)$ reaches the maximum at the depth of 190 m. According to the KdV theory of continuous stratified fluid, the maximum amplitude of internal waves does not occur near the thermocline, but at a depth of about 100 m below the thermocline. The water depth here is about 520 m. If the depth at which the eigenfunction $\varphi$ reaches the maximum value is the interface between the upper and lower water layers of the two-layer fluid KdV model, it is obvious that $h_1 < h_2$, which infers that ISW0729 is a depression wave.

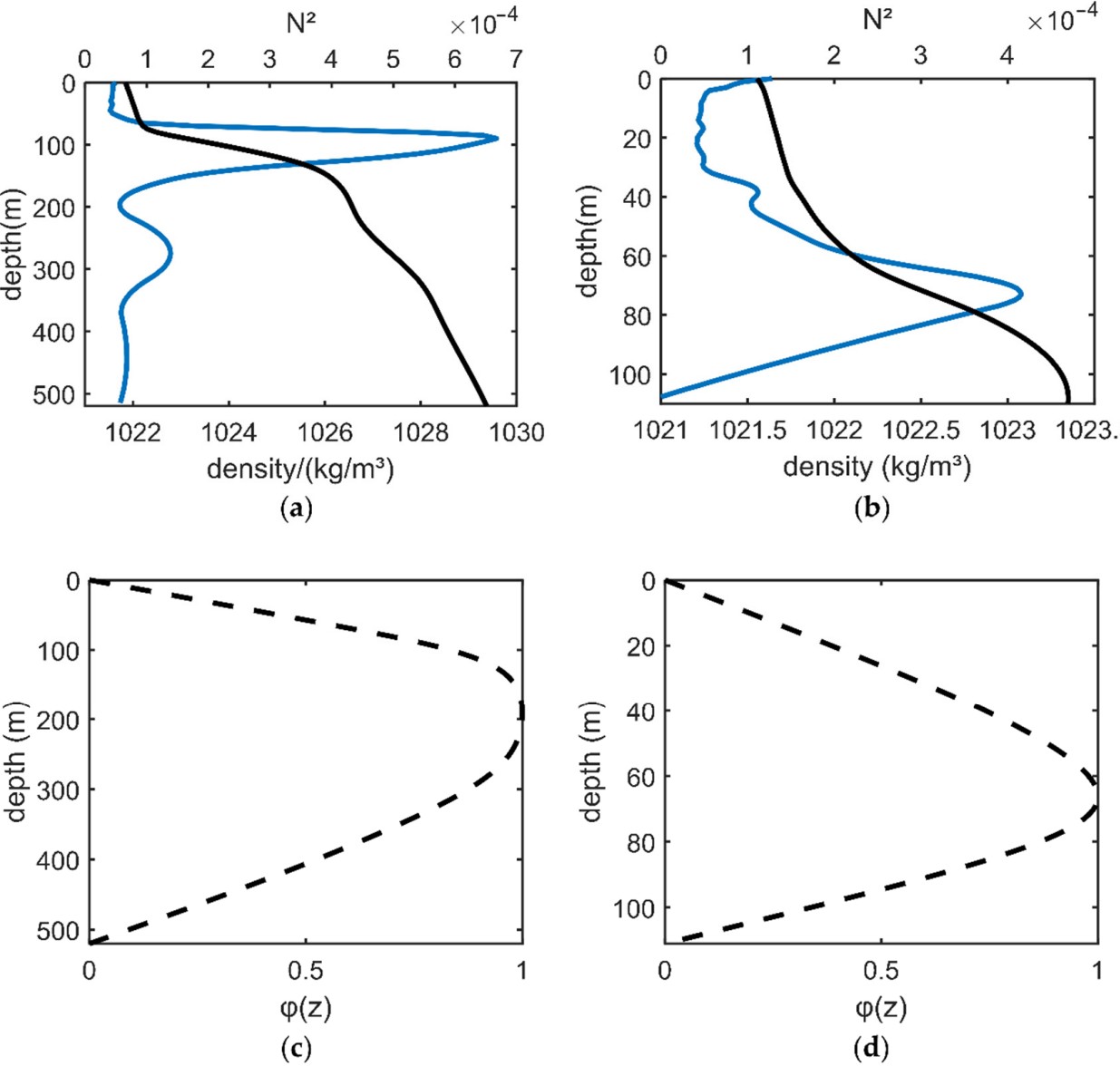

**Figure 6.** (**a**) Climatological potential density and Brunt–Väisälä frequency profiles for ISW0729. (**b**) The first-mode vertical eigenfunction $\varphi(z)$ of ISW0729. (**c**) Climatological potential density and Brunt–Väisälä frequency profiles for ISW0424. (**d**) The first-mode vertical eigenfunction $\varphi(z)$ of ISW0424.

Figure 6b,d show the Brunt–Väisälä frequency and eigenfunction of ISW0424, respectively. The maximum value of the Brunt–Väisälä frequency is at approximately 74 m depth. The eigenfunction $\varphi$ reaches the maximum value at a depth of 67 m, so the maximum depth of the eigenfunction $\varphi$ is in good agreement with the depth of the density profile, which is different from ISW0729. The water depth is 110 m. Regarding the depth at which the eigenfunction $\varphi(z)$ reaches the maximum value as the interface between the upper and lower water layers in the two-layer fluid KdV model, there are $h_1$ = 67 m and $h_2$ = 43 m, respectively. Since $h_1 > h_2$, it can be inferred that ISW0424 is an elevation wave. Since the characteristic half width $L$ and eigenfunction $\varphi(z)$ are obtained, as well as the thickness of upper and lower layers being determined, the parameters of ISWs through a continuously stratified model and two-layer model can be calculated. The main parameters of ISW0729 and ISW0424 are summarized in Table 1.

**Table 1.** Parameters obtained by different reconstruction methods for ISW0729 and ISW0424.

| Name of ISWs | Reconstruction Method | L/m | $\alpha$ | $\beta$ | c/(m/s) | V/(m/s) | $\eta_0$/m |
|---|---|---|---|---|---|---|---|
| ISW0729 | Two-Layer Liquid | 1076 | $-7.4 \times 10^{-3}$ | $2.31 \times 10^4$ | 2.21 | 2.29 | 32.4 |
| | Continuously Stratified Liquid | 1076 | $-1.26 \times 10^{-2}$ | $2.46 \times 10^4$ | 2.04 | 2.13 | 20.2 |
| | Mooring Data | / | / | / | / | 1.92 | 15.7 |
| ISW0424 | Two-Layer Liquid | 205 | $5.9 \times 10^{-3}$ | 275.77 | 0.56 | 0.59 | 18.4 |
| | Continuously Stratified Liquid | 205 | $7.8 \times 10^{-3}$ | 302.50 | 0.52 | 0.56 | 15.3 |
| | Mooring Data | / | / | / | / | 0.47 | 11.1 |

Figure 7 shows the underwater flow field structures of ISW0729 and ISW0424. It can be found that ISW0729 is a depression wave and ISW0424 is an elevation wave, which is consistent with the conclusion of the two-layer KdV model. The maximum horizontal velocity caused by ISW0729 is 27.8 cm/s, and the core with a velocity greater than 20 cm/s exists at the upper part of 100 m. The time for the ISWs passing through a specific position is about 12 min. Below 200 m, the horizontal current direction of ISWs is opposite to the wave propagation direction. In the vertical direction, there are downward and upward currents before and after the wave trough, with the maximum velocity of 1.9 cm/s. The maximum horizontal velocity caused by ISW0424 is 26.2 cm/s, and the core with a velocity greater than 20 cm/s exists at a depth of 85 m near the seabed. The time for the ISWs to pass through a specific position is nearly 7 min. The maximum vertical velocity is 3.8 cm/s, which is twice as large as that of ISW0729.

The KE and APE distribution of ISWs are shown in Figure 8. Since the horizontal flow caused by waves is one order of magnitude larger than the vertical flow, both of the two ISWs' KE are dominated by the horizontal velocity. The maximum APE of ISW0729 is distributed near the density thermocline with a depth of 100 m, rather than at the depth of 190 m with the largest amplitude. Since the density gradient at the thermocline is the largest, where the reduced gravity reaches the maximum, it makes a great contribution to the APE. The maximum value of KE is 39.4 J/m$^3$ at the sea surface, which is equivalent to the maximum value of APE (40.0 J/m$^3$). The total energy obtained by superposition of KE and APE forms a high energy center at a depth of about 90 m, which is just consistent with the depth of density thermocline, and the maximum value of the energy center is 63.8 J/m$^3$. ISW0424 is an elevation wave, which makes its energy distribution show different characteristics from ISW0729. The maximum value of KE is located near the seabed. The maximum APE is also located at the thermocline depth, but its magnitude is only 6.3 J/m$^3$, which is far less than the maximum KE. Due to the large gap between KE and APE, the total energy distribution is mainly dominated by KE, and the high-value center of energy is basically the same as that of KE.

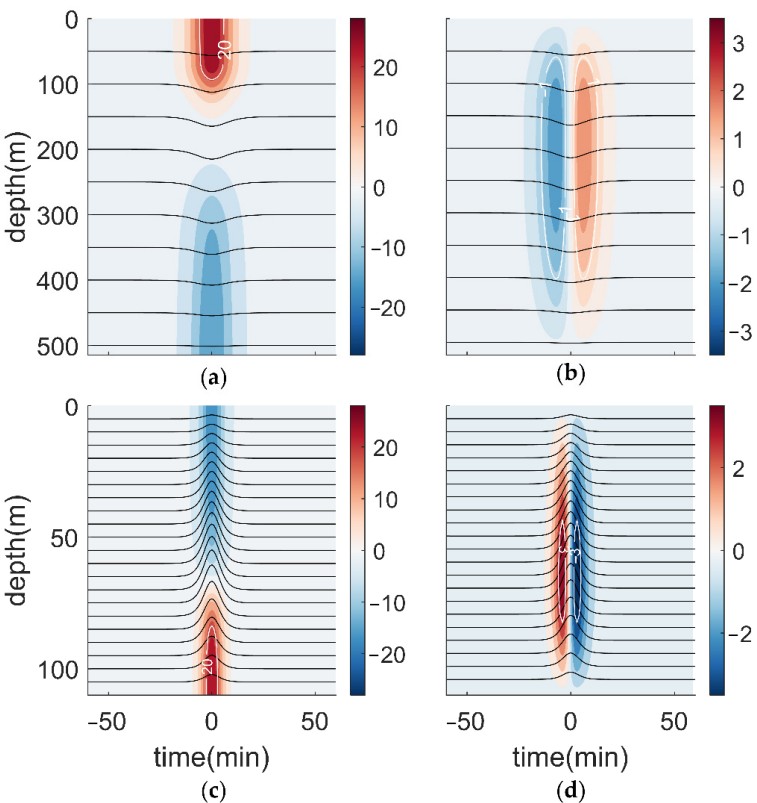

**Figure 7.** Underwater flow field structures of ISWs. (**a**) Horizontal current of ISW0729. (**b**) Vertical current of ISW0729. (**c**) Horizontal current of ISW0424. (**d**) Vertical current of ISW0424.

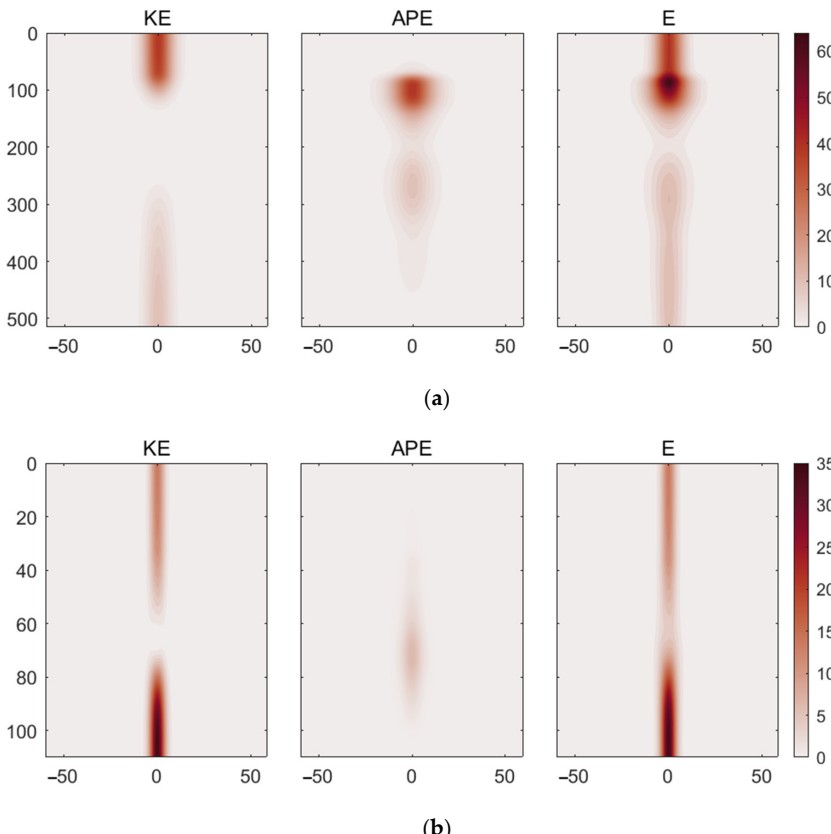

**Figure 8.** Kinetic energy (KE), available potential energy (APE), and total energy (E) of (**a**) ISW0729 and (**b**) ISW0424.

### 3.2.3. Comparison with Moorings Measurement

The reconstruction by the KdV method is verified by the mooring observations. The isotherm lines are used to represent the ISWs as shown in Figure 9. For eliminating the interference of tidal signals in identifying ISWs, the isotherm was run through a high-pass filter for 4 h. The shooting time of SAR image is 21:04:30 on 29 July 2017, when ISW0729 has passed the mooring. The mooring data show that the pilot wave of ISW0729 appears at about 17:30, and the vertical distance between the position of mooring and the peak line on the SAR image is 24.7 km, so the actual wave speed can be calculated as 1.92 m/s. Furthermore, the actual maximum amplitude of ISW0729 is about 15.7 m, which is less than the theoretical amplitude. For the ISW0424 case, the actual wave velocity is 0.47 m/s, which is less than the theoretical prediction. The actual maximum amplitude of ISW0424 is about 11.1 m, which is also smaller than the theoretical amplitude.

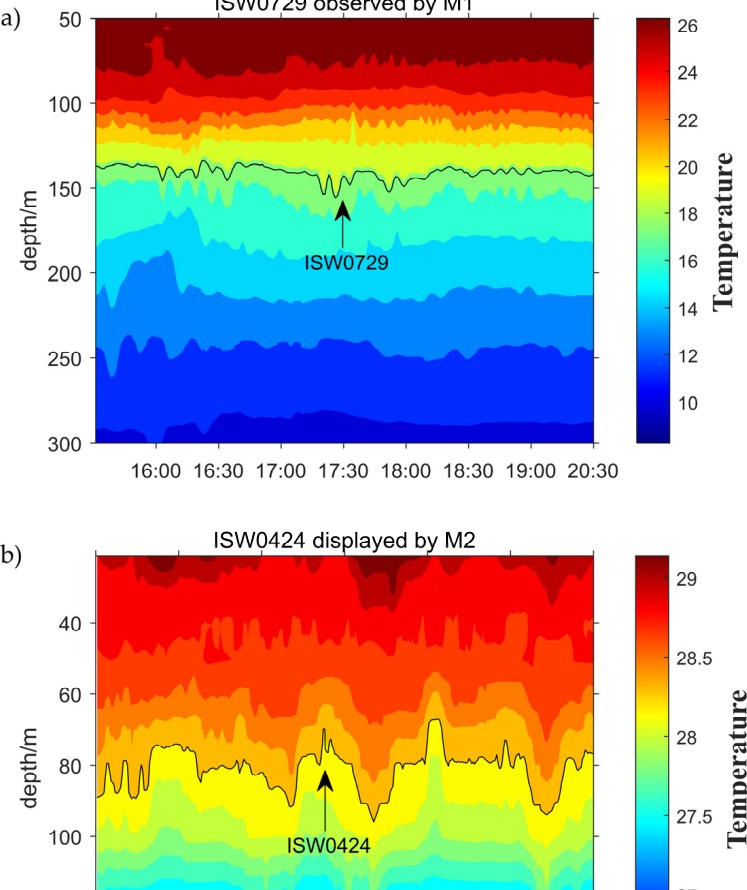

**Figure 9.** Temperature profile measured at the moorings during the ISWs. ISW0729 and ISW0424 are marked by arrows. (**a**) Temperature profile at M1 from 17:00 UTC to 21:00 UTC on 29 July 2017. (**b**) Temperature profile at M2 from 24 April 21:00 UTC to 25 April 03:00 UTC in 2017.

The comparison of the actual amplitude and wave velocity with the results of the two reconstruction models are shown in Table 1. It is found that the amplitude and wave velocity of the two theoretical models are larger, while the error of the continuous layered model is smaller than that of the two-layer model. Therefore, in the Timor Sea, the continuous layered model is more suitable than the two-layer model to describe ISWs. The reconstruction parameters acquired by the continuous layered model are more accurate.

## 4. Discussion

Compared with other hotspots, the distribution of ISWs in the Timor Sea has unique characteristics. As discussed in Section 3.1, compared with other regional seas, such as the Laptev Sea [19] and the White Sea [47], the occurrence frequency of ISWs in the Timor Sea is low. There are many large amplitude ISWs reported in the Lombok Strait [9,10] and northern South China Sea [11,12]. However, there are no comparable amplitude ISWs in the Timor Sea. This may be related to the generation mechanism of ISWs in the Timor Sea, which will be discussed in the following.

It is well known that when the current passes through the undulating terrain, the seawater with density stratification is easily forced by the terrain to gain energy, which leads the generation of ISWs. In the Timor Sea, the Bonaparte basin is a suitable area for the generation of ISWs, which are long and narrow in northeast and southwest directions on the shelf, with a maximum depth of about 170 m. From Figure 10, it seems that most of the ISWs are observed in Bonaparte basin and its edge. It is reasonably inferred that the important generation mechanism of the ISWs in the Timor Sea is the lee wave mechanism. When the steady tidal current flows through the stratified ocean, a lee wave is generated at the back of the raised topography, and gradually evolves into ISWs under the influence of nonlinear effects.

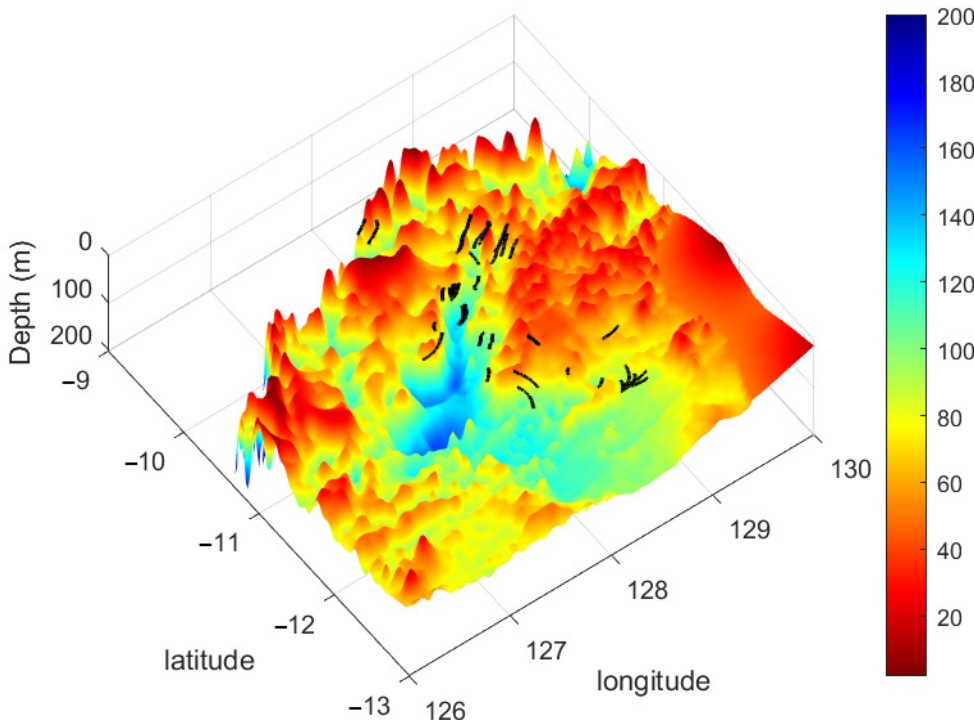

**Figure 10.** ISWs and the three-dimensional seabed topographic map of the continental shelf.

It can be found that the specific topography is the hotspot where ISWs occur (Figure 10). However, the driving mechanisms are still unclear. It is well known that tide is the main energy source for the ISWs. Figure 11 shows the change of barotropic tidal current with time based on FES2014 tidal data, and the time corresponding to the red dotted line is the time when two cases of ISWs appear. It can be seen that both ISW cases are accompanied by high tide peaks during spring tide. Statistical results show that most of the ISWs occurred during spring tide, accounting for 84.1% of the total ISWs, which shows that the spring tide has made a major contribution to the generation of ISWs. Contrastingly, it is difficult to generate ISWs during neap tide, which accounts for 15.9% of total ISWs. Through the above discussion, it can be concluded that the interaction between strong tidal current and basin topography is the main mechanism for the generation of ISWs in the Timor Sea.

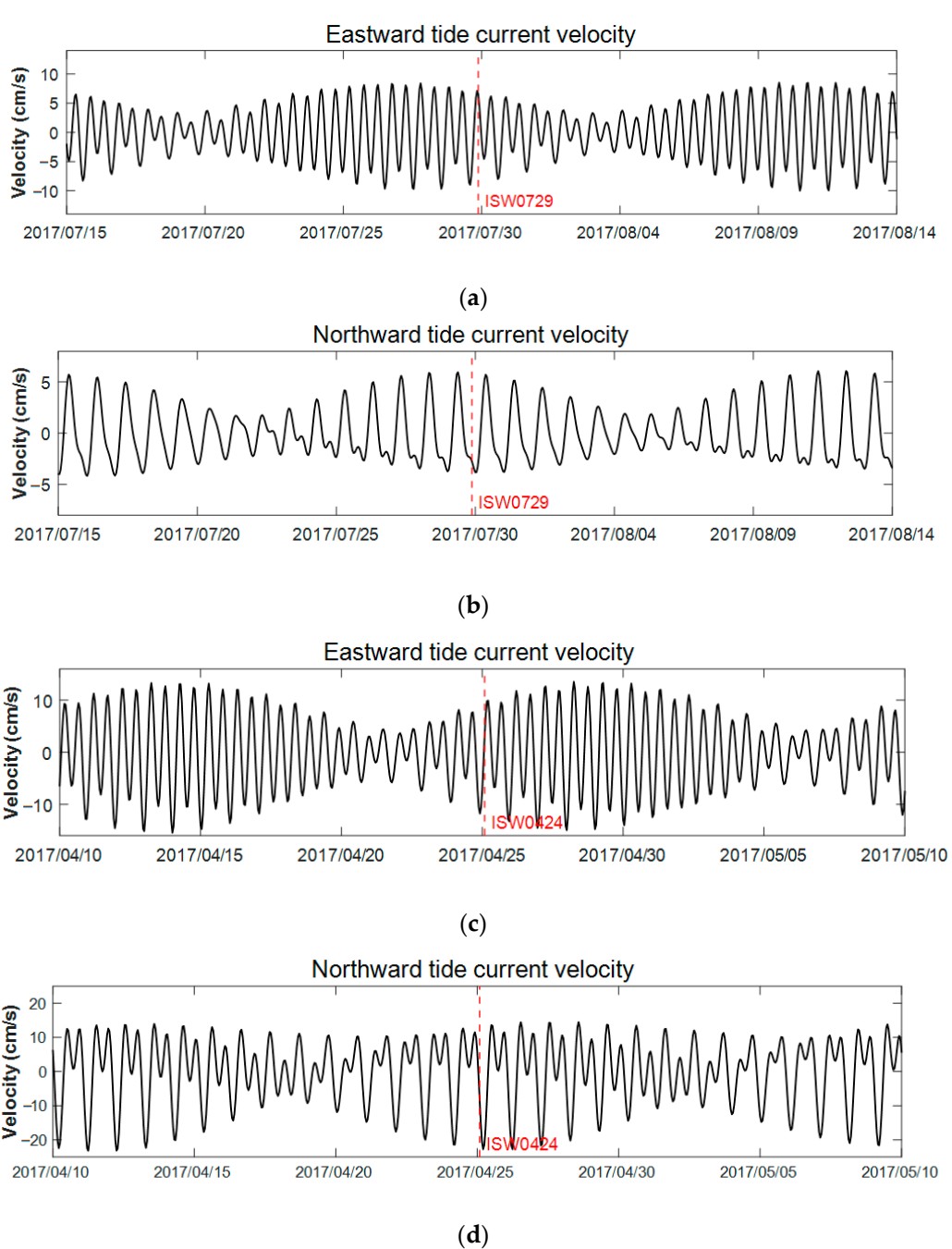

**Figure 11.** Barotropic tide current variation at the location of moorings during ISW cases. The red dotted lines mark the occurrence time of ISWs. (**a**) Eastward tide current variation during ISW0729. (**b**) Northward tide current variation during ISW0729. (**c**) Eastward tide current variation during ISW0424. (**d**) Northward tide current variation during ISW0424.

## 5. Conclusions

Based on the statistical analysis of a large number of remote sensing images, this study explores the temporal and spatial distribution characteristics of ISWs in the Timor Sea. Results show that most of the ISWs appear in Bonaparte basin and in its vicinity on the continental shelf, and only a few appear at the junction of the shelf edge and Timor Trench. The wavelength of most ISWs is less than 400 m, and the waves with the longest wavelength and crest line all appear at the edge of the shelf. The peak line of ISWs is longer in the deeper waters. Most ISWs propagate to the southeast or northwest, which is consistent with the characteristics of local barotropic tidal current. The number of ISWs observed

through remote sensing images is much higher in spring and autumn than in other seasons. However, due to the reason that wind speed is lower in spring and autumn, it is easier to capture ISWs through remote sensing images. Therefore, the seasonal distribution of internal waves in remote sensing images needs to be further validated.

Two typical ISWs are reconstructed based on the KdV equation of two-layer fluid and continuous stratified fluid and are compared with the mooring observations. The results show that, compared with the two-layer model, the parameters of ISWs reconstructed based on the continuous layered KdV equation are more accurate. According to the characteristics of barotropic tide and seabed topography, the formation mechanism of ISWs in the Timor Sea is briefly analyzed. The steep topography and tidal current during spring tide are the two main factors for the formation of ISWs.

ISWs occur widely in all regions of the world's oceans. However, the knowledge of the ISWs' characteristics in the Timor Sea are scarce. This study has improved the understanding of ISWs in this area. However, it is limited by the short observation period and quantity; the detailed characteristics of ISWs have not been fully revealed. In particular, the interaction mechanism between internal tide and topography in the Timor Sea needs to be further elaborated using a numerical model.

**Author Contributions:** Conceptualization, Y.Z. (Yongchui Zhang); methodology, Y.Z. (Yunxiang Zhang) and Y.Z. (Yongchui Zhang); software, Y.Z. (Yunxiang Zhang) and Z.G.; remote sensing image collecting and preprocessing, J.C.; validation, Y.Z. (Yongchui Zhang), M.H. and X.Z.; formal analysis, Y.Z. (Yunxiang Zhang); writing—original draft preparation, Y.Z. (Yunxiang Zhang); writing—review and editing, Y.Z. (Yongchui Zhang); supervision, Y.Z. (Yongchui Zhang) and M.H.; project administration, M.H.; funding acquisition, M.H. and T.X. All authors have read and agreed to the published version of the manuscript.

**Funding:** This research is jointly funded by the Natural Science Foundation of Hunan Province (2023JJ10054), National Natural Science Foundation of China (No. 41875061; No. 41775165; 51609254), the Laoshan Laboratory (No. LSKJ202202700), and the National Natural Science Foundation of China (NSFC) Project (42076024).

**Data Availability Statement:** Sentinel 1 SAR images used in this work were available from the European Space Agency. The Sentinel 1 SAR data can be ordered via https://search.asf.alaska.edu (accessed on 17 December 2022). The mooring data used in this work were sourced from Australia's Integrated Marine Observing System (IMOS) and can be ordered via https://portal.aodn.org.au/search (accessed on 27 December 2022). IMOS is enabled by the National Collaborative Research Infrastructure strategy (NCRIS). The monthly average data of temperature and salinity used to calculate background density profile were obtained from the ocean physics reanalysis product of CMEMS, and the website address of the data is https://data.marine.copernicus.eu/product/GLOBAL_MULTIYEAR_PHY_001_030/services (accessed on 20 January 2023). The barotropic tidal data came from FES2014, which can be downloaded at https://www.aviso.altimetry.fr/en/data/products/auxiliary-products/global-tide-fes.html (accessed on 8 February 2023). FES2014 was produced by Noveltis, Legos, and CLS and distributed by Aviso+, with support from Cnes (https://www.aviso.altimetry.fr/ (accessed on 8 February 2023)).

**Acknowledgments:** We are grateful for Li Huimin at the Nanjing University of Information Science and Technology for her scientific discussions, support with data resources, and valuable opinions throughout this research.

**Conflicts of Interest:** The authors declare no conflict of interest.

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
