# Peer review of "Characteristics of Internal Solitary Waves in the Timor Sea Observed by SAR Satellite"

_remotesensing, doi:10.3390/rs15112878_

Round 1

Reviewer 1 Report

In this paper, the temporal and spatial distribution characteristics of ISWs in the Timor Sea are analyzed by Synthetic Aperture Radar (SAR) images. The study is interesting and valuable. Some questions and suggestions are as follows.

1.      Line 22: “The underwater structures of two typical ISWs are s reconstructed …” Delete the “s”.

2.      Line 30: The concept of internal solitary wave (ISW) can be briefly introduced at the beginning of the Introduction.

3.      Line 165: Provide a detailed explanation of the physical significance and application scenarios of the selected Korteweg de Vries equation.

4.      Illustrate the difference in the temporal and spatial distribution characteristics s of ISW between the Timor Sea and previous research hot areas in Results or Discussions.

5.      Lines 208-217: Repeating with Lines 222-231.

6.      Lines 235-240: The description of the phenomenon does not match Figure 2(b) (c).

7.      Line 248: Are the spatial distribution of ISWs parameters accurate in Figure 2, considering the ISWs in the Timor Sea (Line 107) couldn’t be detected (Line 254-256) effectively?

8.      Line 331: In Table 1, η0/m obtained by different reconstruction methods for ISW0729 and ISW has a significant error.

9.      Lines 439: In Results and Conclusions, from the results shown in Figure 2(d), most ISWs propagate southwest or northeast rather than southeast or northwest.

10.   Lines 493,535,541,542,561 etc. Unify the format of references.

Refine language, simplify long sentences, and improve language expression in this paper.

Reviewer 2 Report

This paper investigates internal waves in the unexplored basin, the Timor Sea, and validates issues resulting from satellite observations of internal waves using buoy measurements. This paper can potentially be published in Remote Sensing, but Section 2 needs more clarification. I recommend a major revision to improve the article.

 General comments

- The empirical mode decomposition procedure must be described in more detail. It must be explained why this method is better than moving averaging along the ISWs propagation direction. What is the area where the empirical mode decomposition was applied? What determines the used numeration of modes? Why is the first mode of the shortest scale? You claimed that the first and second fragments in Figure 5 better fit by the second and third modes, respectively. This contradicts Figure 5. I recommend improving Figure 5 for easy reading. Say, the best modes can be superimposed on the original signals.

- Internal wave solitons in continuously stratified sea must be described in more detail. As the works [36,37] deal only with two-layer sea, please provide reference to the work deriving equations (5,6) and generalizing the solitons theory for continuously stratified water. Particularly it needs to explain why linear theory, say equation (7), is applied to the nonlinear soliton problem. Why do you use linear equation (15), omitting the term ?  Also, the calculation of eigenfunctions must be clarified in subsection 3.2.2.

 Specific comments

- L129-130 Please clarify the sentence: “There are 345 SAR images totally with the path of 119 and the frame of 125 and 130 during the six years from 2017 to 2022”

- L 203 Why do you do integration over horizontal ranges of ISW in deriving the equation (17)?

- Eq (17) The term needs a definition.

 - Lines 222-231 repeat the preceding paragraph

- L290-291, “the half wave width of ISW0729 could be calculated as 1076 m according to equation (8).” – Equation (8) yields the amplitude of the soliton, not the wave width. Please clarify it.

 - L298 Replace “fifteen” with “fifth”.

 - Figure 5 – Why are the fifth panels labeled “Res”? Please correct it.

 - L309 “the maximum amplitude of internal waves does not occur near the thermocline, but at a depth of about 100 m below the thermocline” – Why is it 100 m? It must depend on the water density profile. Please clarify it.

 - L559 – Incomplete reference [34]. Please correct.

Round 2

Reviewer 2 Report

The authors improved the manuscript according to all my comments. This version of the paper can be accepted for publishing. However, I suspect checking English is needed. Also, several “Error! Reference source not found” are presented in the manuscript.